

# Multi-modal sleep intervention for community-dwelling people living with dementia and primary caregiver dyads with sleep disturbance: protocol of a single-arm feasibility trial

Sumedha Verma[1], Prerna Varma[1], Aimee Brown[1,2], Bei Bei[1], Rosemary Gibson[3], Tom Valenta[4], Ann Pietsch[5], Marina Cavuoto[1], Michael Woodward[6], Susan McCurry[7] and Melinda L. Jackson[1]

[1] Monash University, Clayton, Victoria, Australia
[2] Monash-Epworth Rehabilitation Centre, Richmond, Victoria, Australia
[3] Health and Ageing Research Team, School of Psychology, Massey University, Palmerston North, New Zealand
[4] Independent Researcher, Melbourne, Victoria, Australia
[5] Independent Researcher, Adelaide, South Australia, Australia
[6] Aged and Continuing Care Services, Austin Health, Heidelberg, Victoria, Australia
[7] School of Nursing, University of Washington, Washington, United States of America

Corresponding author
Melinda L. Jackson,
melinda.jackson@monash.edu

## ABSTRACT

**Background.** Disturbed sleep is common among people living with dementia and their informal caregivers, and is associated with negative health outcomes. Dyadic, multi-modal interventions targeting caregiver and care-recipient sleep have been recommended yet remain limited. This protocol details the development of a single-arm feasibility trial of a multi-modal, therapist-led, six-week intervention targeting sleep disturbance in dyads of people living with dementia and their primary caregiver.
**Methods.** We aim to recruit 24 co-residing, community-dwelling dyads of people living with dementia and their primary informal caregiver ($n = 48$) with sleep concerns (Pittsburgh Sleep Quality Index ≥5 for caregivers, and caregiver-endorsed sleep concerns for the person living with dementia). People who live in residential care settings, are employed in night shift work, or are diagnosed with current, severe mental health conditions or narcolepsy, will be excluded. Participants will wear an actigraph and complete sleep diaries for two weeks prior, and during the last two weeks, of active intervention. The intervention is therapist-led and includes a mix of weekly small group video sessions and personalised, dyadic sessions (up to 90 min each) over six weeks. Sessions are supported by a 37-page workbook offering strategies and spaces for reflections/notes. Primary feasibility outcomes are caregiver: session attendance, attrition, and self-reported project satisfaction. Secondary outcomes include dyadic self-reported and objectively-assessed sleep, depression and anxiety symptoms, quality of life, and social support. Self-report outcomes will be assessed at pre- and post-intervention.
**Discussion.** If feasible, this intervention could be tested in a larger randomised controlled trial to investigate its efficacy, and, upon further testing, may potentially

represent a non-pharmacological approach to reduce sleep disturbance among people living with dementia and their caregivers.

**ANZCTR Trial registration**. ACTRN12622000144718: https://www.anzctr.org.au/Trial/Registration/TrialReview.aspx?id=382960&showOriginal=true&isReview=true

# INTRODUCTION

People living with dementia commonly experience changes to their sleep including frequent nighttime awakenings, daytime sleepiness and napping, changes in circadian functioning and "sundowning"—periods of agitation and disorientation commonly during the late afternoon and early evening (*Elder et al., 2022*; *Wennberg et al., 2017*; *Ancoli-Israel et al., 1994*). Some of these changes can be challenging, with over 70% of people living with dementia experiencing at least one type of sleep disturbance (*Rongve, Boeve & Aarsl, 2010*) including insomnia (*Wennberg et al., 2017*; *Mayer et al., 2011*), which is characterised by significant difficulties in initiating or maintaining sleep which cause distress and/or impact daytime functioning (*American Psychiatric Association, 2013*). There also is growing evidence that sleep disturbances and sleep disorders, such as insomnia, can increase the risk of developing all-cause dementia (*Shi et al., 2018*). Thus, treatment of poor sleep may be an important early prevention strategy for reducing the likelihood of future cognitive decline.

Sleep disturbances not only affect people living with dementia but those who support them at home (*Gao, Chapagain & Scullin, 2019*). In Australia, 91% of community-dwelling people living with dementia are informally supported by caregivers at home, many of whom are family members that provide 40 hours or more of informal care per week (*Brooks, Ross & Beattie, 2015*). While there is growing research into the positive aspects of caregiving (*Quinn & Toms, 2019*), caregivers commonly experience poor sleep. Over 91% of caregivers report poor sleep quality (*Peng, Lorenz & Chang, 2019*; *Smyth et al., 2020*), and one meta-analysis and systematic review found that caregivers reported significantly less sleep (up to 3.5 h loss per week) and had poorer sleep quality than non-caregiving age-matched peers (*Gao, Chapagain & Scullin, 2019*). Insomnia represents another concern among caregivers (*McCurry, Song & Martin, 2015*). In a recent study, 55% of caregivers report sleep onset latencies of over 30 min, with almost 20% who report times of over 60 min to fall asleep at night (*Smyth et al., 2020*). As the number of caregivers is projected to significantly rise over coming years (*Brooks, Ross & Beattie, 2015*), supporting unpaid caregivers through accessible ways represents an important area of research focus.

The underlying causes of sleep characteristics, such as poor sleep quality, in caregivers supporting people living with dementia are multifactorial. Caregiver demographic and psychosocial factors, as well as sleep-wake behaviours in the person they support can all

increase the risk of disturbed sleep (*Peng, Lorenz & Chang, 2019*; *Peng & Chang, 2013*). For instance, poorer caregiver sleep quality has been associated with greater symptoms of caregiver stress, depression, and anxiety (*Peng, Lorenz & Chang, 2019*; *Smyth et al., 2020*; *Van Hout et al., 2022*) and predicts greater levels of fatigue (*Chang et al., 2020*). Additionally, sleep disturbances in people living with dementia have been strongly associated with greater levels of overload in caregivers (*Gehrman et al., 2018*), and are a key predictor of transition to formal residential care (*Hope et al., 1998*). This highlights the complex interplay of caregiver and care recipient sleep and wellbeing, which hold broader systemic implications.

Past studies have described how poor sleep in caregivers may be precipitated and perpetuated by sleep disturbances in the person they care for, such as tending to nighttime awakenings or wandering, but may not be fully explained by care recipients' sleep (*McCurry, Song & Martin, 2015*; *McCurry et al., 2008*). Caregivers may engage in unhelpful sleep practices to compensate for disturbed nighttime sleep and daytime fatigue, such extending time in bed, daytime napping or using stimulants, which may consequently perpetuate experiences of poor sleep (*Peng, Lorenz & Chang, 2019*; *McCurry, Song & Martin, 2015*). Sleep disturbances in caregivers may also be maintained by stress, depression, and limited understanding of good sleep practices (*Gao, Chapagain & Scullin, 2019*; *McCurry et al., 2007*). From a cognitive behavioural perspective, this may place caregivers at a higher risk of developing and maintaining insomnia (*McCurry, Song & Martin, 2015*). Given the unique vulnerabilities to chronic sleep disturbance among those supporting people living with dementia at home, interventions that promote caregiver sleep health and daytime functioning are imperative, and may in turn improve the wellbeing and quality of care provided to those being supported.

Several interventions hold potential for improving sleep in caregivers and people with dementia (Table 1). Cognitive behavioural therapy (CBT) for insomnia is a non-pharmacological intervention that addresses unhelpful thoughts and behaviours associated with poor sleep; it improves sleep quality in general caregivers (*Cooper et al., 2022*; *Fernandez-Puerta, Prados & Jimenez-Mejias, 2022*), caregivers of people living with dementia (*Gao, Chapagain & Scullin, 2019*; *Pignatiello et al., 2022*) and people with mild cognitive impairment (*O'Caoimh et al., 2019*). Mindfulness-based interventions involve relaxation strategies and moment-to-moment awareness, which have been associated with reduced insomnia symptoms in caregivers (*Jain, Nazarian & Lavretsky, 2014*) and improved quality of life and mood in people with dementia (*Hoffman et al., 2019*). Light therapy involves appropriately-timed exposure to bright light to realign the circadian rhythm with externally required sleep/wake timing (*Terman & Terman, 2011*) and/or to reduce daytime sleepiness (*Phipps-Nelson et al., 2003*). Light therapy has been associated with significant improvements in circadian-related outcomes and sleep quality in caregivers (*Pignatiello et al., 2022*), with some sleep improvements in people living with dementia (*O'Caoimh et al., 2019*; *Sidani et al., 2022*). Similarly, physical activity interventions have been significantly associated with reduced sleep latency and greater sleep quality in caregivers (*Pignatiello et al., 2022*) and are associated with improved functional capacity in

**Table 1  Sleep and circadian interventions for caregivers and persons living with dementia.**

| Intervention | Description | Summary of findings |
|---|---|---|
| Cognitive Behavioural Therapy for Insomnia | A multi-component intervention that addresses unhelpful thoughts and behaviours associated with poor sleep Incorporates sleep education and sleep hygiene, combined with behavioural and cognitive components | Caregivers: Improvements in sleep quality (*Cooper et al., 2022*; *Fernandez-Puerta, Prados & Jimenez-Mejias, 2022*)<br><br>Person living with dementia: Improvements in actigraphy-assessed sleep parameters (*Poon, 2022*) |
| Mindfulness interventions | Meditation training, relaxation strategies and moment-to-moment awareness | Caregivers: Reduced insomnia symptoms; improvements in sleep quality (*Pignatiello et al., 2022*; *Jain, Nazarian & Lavretsky, 2014*)<br><br>Person living with dementia: Improved quality of life and mood (*Hoffman et al., 2019*) |
| Light therapy | Bright light therapy (BLT):<br>• Involves the use of a light box placed at eye level, one meter in front of the patient.<br>• Recommended dosage between 20 min to two hours in the morning.<br>• BLT is conducted at least 5 days a week, over 10 days to two months, notably in winter times.<br><br>Natural outdoor light exposure:<br>• Spend time outdoors by walking or sitting during the recommended exposure time, or<br>• Sit next to a window with open blinds during the recommended exposure time.<br>• Recommended exposure time is 1 h in the morning around 9 a.m. | Caregivers: Improvement in circadian outcomes and sleep efficiency (*Livingston et al., 2019*; *Song et al., 2021*)<br><br>Person living with dementia: Some evidence of improvements in objective and subjective sleep measures; but mixed findings across studies (*Calvert et al., 2018*)<br><br>Small to moderate effect on circadian amplitude and night time awakenings/ sleep quality. Minimal effect on sleep duration and sleep efficiency (*Sidani et al., 2022*) |
| Physical activity interventions | Provide suggestions for engaging in daytime activities (*e.g.*, gardening, walking)<br><br>Structured activities (*e.g.*, Tai Chi, resistance training) | Caregivers: Reduced sleep latency, improved sleep quality (*Pignatiello et al., 2022*)<br><br>Person living with dementia: Improved functional capacity (*Meyer & O'Keefe, 2018*) |

people living with dementia (*Meyer & O'Keefe, 2018*). Each of these interventions seem to have differential effects on sleep and daytime functioning (Table 1).

Recent systematic reviews have recommended *multi-modal* interventions that incorporate different therapeutic components (*i.e.*, cognitive and behavioural strategies, light therapy, exercise, mindfulness and relaxation) to comprehensively target determinants of sleep disturbance in caregivers and people with dementia (*Pignatiello et al., 2022*; *O'Caoimh et al., 2019*). Some studies have adopted a multi-modal approach to treating sleep disturbance in people living with dementia (see *Poon (2022)* for a review of these interventions and evidence of their efficacy) (*Sidani et al., 2022*; *Livingston et al., 2019*). In addition, *dyadic* interventions that jointly addresses poor sleep in both caregivers and care recipients have been recommended (*Fernandez-Puerta, Prados & Jimenez-Mejias, 2022*; *Pignatiello et al., 2022*). Dyadic interventions, in which both the caregiver and care-recipient are simultaneously targeted (*Poon, 2022*), present unique opportunities to concurrently target poor sleep in people living with dementia and those who informally support them. For example, caregivers may learn to facilitate healthy sleep practices in those

they support while targeting their own sleep in tandem, thereby addressing precipitating and perpetuating factors of disturbed sleep more comprehensively.

In-home dyadic interventions may also be cost-effective; demands on external services may be significantly reduced given that sleep disturbance and daytime impacts are managed predominantly at home. However, with few exceptions (*Song et al., 2021*), efficacious, multi-modal, *dyadic* interventions for caregivers and people living with dementia with sleep disturbance remain limited.

The current article presents a protocol of a feasibility trial of a multi-modal dyadic intervention that is personalised, manualised, and group-based to target sleep disturbance in caregivers and people living with dementia. The intervention combines strategies from *cognitive-behavioural* and *mindfulness-based* strategies, *light therapy,* and *physical activity* for caregivers to address their own sleep concerns and support healthy sleep practices in the person they care for. The trial aims to assess program feasibility for caregivers (attendance and attrition rates, and satisfaction using Bowen criteria), and preliminary efficacy for improving sleep, quality of life, and wellbeing outcomes in the dyad.

## METHODS

### Design

This is a single-arm feasibility trial (Fig. 1). Self-reported outcomes will be assessed at pre-intervention (T1, Week 0; within one week of intervention start date), post-intervention (T2; Week 6), and a Week 10 follow-up (T3). The primary endpoint is feasibility at T2. This protocol follows SPIRIT-PRO (*Calvert et al., 2018*) and TIDieR (*Hoffmann et al., 2014*) guidelines (see Supplementary). Due to the nature of the trial design, neither participants nor researchers will be blinded. Ethics approval was obtained by the Monash University Human Research Ethics Committee (30710).

### Participants

Participants will be community-dwelling, co-residing dyads of people living with dementia and their primary caregiver (see Table 2 for how caregivers were operationalised) in Australia and New Zealand. Table 2 lists inclusion and exclusion criteria. Participants will not be restricted in receiving any usual care for health (including mental health) conditions. Given the feasibility nature of the current study and the prevalence of medication usage in our target population, we will not exclude participants taking medication known to affect sleep. Broad inclusion criteria were set due to the feasibility nature of the trial; assessment of characteristics of responders (*i.e.,* type/severity of dementia diagnosis) and other considerations may be undertaken to aid the development of a future randomised controlled trial.

### Procedure
#### Recruitment
The trial is referred to as the "Dementia, Sleep and Wellbeing Study" and will involve varied recruitment methods. Information about the trial will be made available on the Dementia Australia and StepUp for Dementia Research websites. Dementia and caregiver support
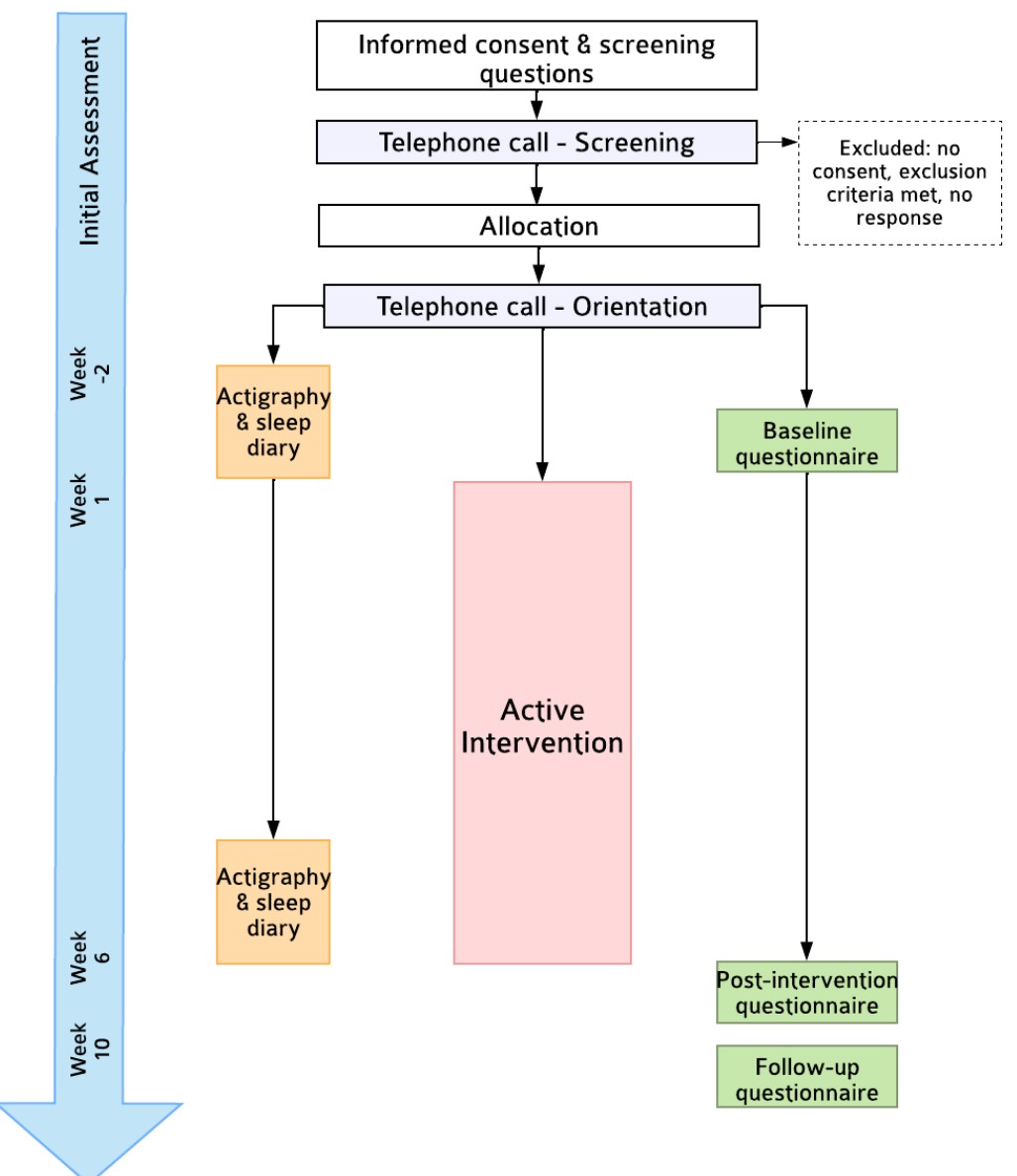

**Figure 1  Flowchart of trial procedures.**

groups will be approached to share information with members. Project information and the accompanying link to the online explanatory statement is posted on Facebook caregiver and dementia support groups in Australia and New Zealand. Flyers are also placed in community centres and medical practices.

### Informed consent

Recommendations from the UK Alzheimer's Society (*Alzheimer's Society, 0000*) for obtaining informed consent from people living with dementia will be followed. The link to the explanatory statement first asks potential participants to select one of two options ("I

**Table 2 Inclusion and exclusion criteria at allocation.**

|  | People living with dementia | Caregivers |
|---|---|---|
| Inclusion | 1. Living with dementia (any type)<br>2. Receive overnight support from a caregiver at least three nights per week<br>3. Ability to provide informed consent, or provide assent to their caregiver<br>4. Endorsed sleep concerns as rated "*Yes*" on a single-item[*] by their primary caregiver | 1. Living with and supporting a person with dementia at least three nights per week<br>2. Ability to communicate in English<br>3. Regular access to phone, email, and internet<br>4. Obtain scores of five or more on the Pittsburgh Sleep Quality Index |
| Exclusion | a) Living in a residential care home<br>b) Have severe, untreated medical or physical health conditions that directly and significantly impact sleep (*e.g.*, severe pain, emphysema)<br>c) High risk of harm to self/others | a) Report a diagnosis of schizophrenia, bipolar disorder or posttraumatic stress disorder<br>b) Have severe, untreated medical or physical health conditions that directly and significantly impact sleep (*e.g.*, severe pain, emphysema)<br>c) Undertake fixed night shift work between midnight and 5am, or rotating work schedules that require night shifts during participation<br>d) Report a diagnosis of narcolepsy<br>e) High risk of harm to self/others |

Notes.

[*]The single item assessing sleep concerns in people living with dementia is: "*Do you think that sleep is an issue in the person you're supporting?*".

am a person living with dementia" or "I am a carer for someone living with dementia") after which they will be directed to the appropriate explanatory statement (*i.e.,* one for people with dementia or one for caregivers). This will include information about assessments, the intervention, research procedures, potential risks, reimbursement, confidentiality and freedom to withdraw. After reading the explanatory statement, participants will complete a consent form (see Supplementary) and provide contact details. Caregivers will be asked whether they believe the person they are supporting will be able to provide informed consent or assent during initial screening questions (detailed below). Consent/assent will be re-confirmed during a telephone call with both the caregiver and person living with dementia.

### *Screening and risk assessment*

Following consent, participants will be directed to an online form to complete brief screening questions which assesses initial eligibility criteria (*e.g.,* for caregivers: "*Do you have current mental health conditions [e.g., schizophrenia, bipolar disorder, PTSD] that directly impact sleep?*"). If exclusion criteria have not been met, participants will be contacted *via* email to schedule a telephone call. Participants who meet exclusion criteria following screening questions will be contacted to confirm ineligibility.

In the screening telephone call, author AB, a provisional psychologist, will: (a) explain the purpose and details of the project and intervention; (b) confirm participants' interest to participate; (c) assess caregiver sleep disturbance *via* Pittsburgh Sleep Quality Index (PSQI; *Buysse et al., 1989*) and sleep disturbance in the person with dementia through the Sleep Disorders Inventory (SDI; *Tractenberg et al., 2003*); (d) provide details of questionnaires (*i.e.,* format and expected duration of roughly 15–20 min); (e) discuss capacity of the person with dementia to complete questionnaires (people with dementia will be encouraged to

complete assessment materials if they, or their caregiver, report capacity to do so) and; (f) conduct a risk assessment.

The risk assessment is undertaken to assess any risk of harm to self and others (*i.e.,* suicidal/homicidal ideation, plan, intent, and previous risk behaviours). Those deemed as high risk are referred to appropriate clinical mental health services and are ineligible to participate; any risk-related issues are discussed with authors SV and MJ (both psychologists), and an external clinical psychologist (described later). Those deemed ineligible to participate after the screening telephone call will be provided a comprehensive list of external support services and excluded from the trial.

### *Intervention*

After screening, eligible participant dyads will participate in a 20-minute "pre-orientation" session *via* video with a trained research assistant to explain use of actigraph and sleep diary, which are sent *via* post prior to predetermined start date. Participants will complete an online baseline questionnaire within a week prior to beginning the intervention; automatic prompts to complete all items will be provided, and email reminders will be sent if questionnaires have not been completed within five days of release. Eligible participants will receive the intervention as detailed below. Participants will wear an actigraph and complete sleep diaries daily for two weeks prior to intervention start date, and then return the watches and diaries to the research team to be downloaded and recharged. The actigraphs and new diaries will be posted to be worn again during Weeks 5 and 6 of the intervention. A trained research assistant will manage actigraph recharging and monitor data. Sleep diaries and actigraphs are returned following completion of the follow-up questionnaire (Week 10). Ethics approval will be sought by Monash University Human Research Ethics Committee for any protocol modifications before being implemented and communicated to participants if appropriate, and will be documented on a running sheet.

## Community and consumer involvement

Authors AP and TV are community advisors with lived experience of dementia and caregiving respectively, and hold vast experience and expertise in advisory and research collaboration. Community advisors are heavily involved in all stages of project development and provide input on (but not limited to): explanatory statements and consent forms, study design, informed consent procedures, development and delivery of focus groups and interviews, intervention materials, language recommendations, questionnaire content and aesthetics, recruitment methods, promotional materials, and other consumer-facing content.

In late 2021, two focus groups (one for caregivers and one for people with dementia [$n = 3$ each group]), and five semi-structured interviews were undertaken to: (a) identify common sleep and daytime changes; (b) identify helpful/unhelpful strategies for sleep and daytime effects; (c) explore barriers to help-seeking; (d) seek feedback and recommendations on proposed study design, intervention content, logistics, and predicted feasibility. Qualitative and quantitative feedback arising from focus groups and interviews were documented in a tracking sheet and incorporated in the development of the intervention. Suggestions

included a mix of dyadic and small group delivery, weekly sessions of 90 min duration, focusing content on resourcing caregivers, and making it optional for the person living with dementia to attend (manuscript currently in preparation; A Brown, J Dowling, S Verma, B Bei, R Gibson, A Piestch, T Valenta, M Cavuoto, M Woodward, S M McCurry, M L. Jackson, P Varma, 2023, unpublished data).

## Intervention

The intervention delivery has two components: (1) weekly therapist-led personalised and group sessions delivered *via* video conferencing over six weeks; and (2) a hardcopy 37-page participant workbook (see Supplementary). The workbook includes a reflective journal and contains a list of resources to support participants' learning and application of new skills and promote adherence.

The intervention will be delivered by registered psychologists with expertise in the provision of sleep interventions under the supervision of a clinical psychologist with experience working with people living with dementia and caregivers. The lead intervention developer (SV) holds experience in the design, development and delivery of sleep-based interventions, and undertook additional training in dementia awareness, and caregiver sleep and wellbeing to assist content development. Manualised scripts for each session were developed to ensure consistency.

The intervention content is multi-modal and draws from CBT for insomnia, mindfulness, physical activity interventions and light therapy. Different components include: collaborative goal-setting, psychoeducation, practical sleep strategies based on stimulus control and sleep hygiene, relaxation strategies (*e.g.*, deep breathing, mindfulness), physical activity recommendations (*e.g.*, walking, gardening, stretching), gaining natural light during the morning/daytime and avoiding bright light at night, prioritising rest and self-care, enlisting support from others, and strategies to manage fatigue (*e.g.*, planning, pacing, taking breaks).

The intervention adopts a "toolkit" approach based on feedback arising from community consultation, in which dyads learn of several strategies and select to use those that suit their unique, and often changing, circumstances. Intervention providers take a validating, non-judgemental, supportive, and motivational stance to encourage and empower participants to make changes in their everyday lives; these therapeutic qualities were identified through community consultation as being particularly salient given the unique needs of the study population. The intervention components were designed to upskill caregivers in addressing their own sleep and daytime concerns while supporting sleep in the person they care for, however some components (*e.g.*, relaxation, physical activity) were designed to concurrently apply to both caregivers and people living with dementia. Caregivers will be invited to attend all sessions with optional attendance for people living with dementia (except Session 5 which is offered specifically to caregivers following recommendations from community consultation).

Participants will attend six sessions in total (see Table 3). All sessions will run for up to 90 minutes except for Session 4 (45 minutes). Sessions 1 and 4 will involve individual dyads while remaining sessions (*i.e.,* Sessions 2, 3, 5 and 6) will be delivered in small groups

with no more than four dyads (or caregivers in Session 5). Session 1 will be framed as an "orientation" to the project which aims to: (a) identify current sleep concerns and collaborate on personalised goals for the project; (b) provide participants psychoeducation on sleep and insomnia, and provide key sleep strategies; (c) offer recommendations based on identified goals and; (d) discuss ways of overcoming barriers to encourage adherence, including use of the workbook. Participants receive all core components of the intervention however components specifically relevant to the dyad in light of their unique circumstances and goals are highlighted, thus *personalising* the intervention. For example, relaxation strategies may be discussed for a dyad in which sleep onset difficulties are a reported priority, or fatigue management strategies for another dyad that identifies the goal to feel less fatigued during the day.

Group sessions will generally include: (a) an initial review of the previous session and time to answer questions; (b) psychoeducation of key concepts and strategies specific to each session and; (c) guided discussion to enable sharing and collaboration, and reference to relevant sections of the workbook. The purpose of Session 4 is to provide a personalised check-in to encourage engagement and adherence, answer any questions and collaborate a plan for the remaining period of intervention.

Group guidelines will be discussed during Sessions 1 and 2 and reiterated in the workbook (See Supplementary). Community advisors will be invited to attend sessions to provide recommendations for intervention providers to promote respectful and sensitive delivery. All sessions are video-recorded and offered to dyads unable to attend sessions, and for researcher training purposes.

## Measurements

Attendance and attrition rates will be recorded. Various measures of sleep, mood, daytime functioning, quality of life, social support, and other factors will be assessed, either through online questionnaires *via* Qualtrics software (*Qualtrics, 2018–2020*) or hard copies posted to participants, to measure preliminary efficacy (Table 4). Participants will be asked whether they are completing the questionnaires for themselves, or, for caregivers, on behalf of the person they care for at the start of each questionnaire. The post-intervention (T2) questionnaire will gather quantitative/qualitative feedback on intervention helpfulness and usefulness, and assess the preliminary efficacy based on secondary outcomes. The 10-week follow-up (T3) will assess only secondary outcomes.

### Primary outcome

Feasibility will be assessed using domains outlined by *Bowen et al. (2009)*. Implementation will be assessed by program adherence (defined as the number of sessions attended) and drop-out rate. Acceptability will be assessed *via* self-report of the overall program satisfaction and perceived benefit using the Client Satisfaction Questionnaire at post-intervention (CSQ; *Attkisson & Zwick, 1982*) and responses from the post-study evaluation survey. Limited efficacy will be assessed through measures outlined below.

**Table 3   Outline of personalised and group sessions.**

| Session 1 | Introduction (Individual dyads) |
|---|---|
| | Acknowledgement of Country |
| | Overview of project |
| | Psychoeducation: |
| | • Typical sleep changes in people living with dementia and caregivers |
| | • Sleep deprivation vs insomnia |
| | • ''3 ingredients for good sleep''—sleep drive, circadian rhythm, quiet mind |
| | Cognitive behavioural strategies for insomnia: |
| | • Stimulus control (*e.g.*, only going to bed when sleepy, getting out of bed if stimulated and unable to sleep) |
| | • Anchoring wake time |
| | • Guidance on napping |
| | • Sleep hygiene |
| | Collaborative goal-setting, motivational interviewing, personalising strategies |
| | Workbook exercises |
| **Session 2** | **Light, Night and Exercise (Group Session)** |
| | Introduction (including Acknowledgement of Country, group guidelines) |
| | Psychoeducation: |
| | • Circadian rhythm and sundowning |
| | • Role of light and exercise on circadian rhythms (including blue light) |
| | • Effect of light at different times of day/night |
| | Light therapy strategies: |
| | • Natural light exposure upon awakening and during daytime |
| | • Avoidance of bright lights at night (using lamps, dim lights) |
| | • Guidelines on electronic device use |
| | Psychoeducation: |
| | • Exercise and sleep, physical and mental health |
| | Encouraging physical activity (*i.e.,* 30 min of physical activity a day) |
| | Group discussion (implementing light and night strategies into daily routine) |
| | Workbook exercises |
| **Session 3** | **Self-care and Relaxation (Group Session)** |
| | Brief Q&A time |
| | Psychoeducation: |
| | • Self-care |
| | • Normalising stress and worries in caregivers and people living with dementia |
| | Enlisting formal/informal supports |
| | Group discussion (identifying relaxation strategies) |
| | Mindfulness: |
| | • Experiential deep breathing and body scan |
| | Relaxation strategies: |
| | • Worry journal |
| | • Self-talk |
| | Identifying prompts/cues to encourage adherence (*e.g.*, hand-written notes) |
| | Guidance for nighttime awakenings (for caregivers) |
| | Group discussion (implementing relaxation exercises into routine, reminders) |
| | Workbook exercises |

**Table 3** (*continued*)

| Session 1 | **Introduction (Individual dyads)** |
|---|---|
| Session 4 | **Check In (Individual dyads)** |
| | Discussion of changes in sleep and wellbeing |
| | Review of goals set in Session 1 |
| | Identifying helpful/unhelpful strategies |
| | Identifying and collaboratively addressing barriers to engagement |
| | Personalised planning and goal-setting |
| Session 5 | **Managing Fatigue (Group Session—Caregivers only)** |
| | Brief group discussion of relaxation strategies |
| | Psychoeducation: |
| | • Fatigue and components |
| | • Normalising fluctuations in energy levels |
| | • Fatigue vs sleepiness |
| | Self-monitoring energy levels (mobile phone analogy) |
| | Fatigue management strategies: |
| | • Plan, pace and prioritise tasks |
| | • Taking breaks |
| | • Keeping realistic expectations |
| | • Enlisting social support |
| | • Engaging in a 'daily joy' |
| | Group discussion (self-monitoring, identifying and addressing potential barriers to engagement) |
| Session 6 | **Review and Beyond (Group Session)** |
| | Review of previous session content |
| | Group discussion (reflections of project, strategies and challenges) |
| | Relapse prevention: |
| | • ''Traffic light'' analogy (green, amber and red zones of wellbeing) |
| | • Self-monitoring and identifying early warning signs |
| | • Contingency planning for challenging times |
| | Group discussion—self-monitoring, planning |
| | Thank you and farewell |

### Secondary outcomes

1. *Sleep quality* measured by the PSQI, which has adequate reliability and validity in community samples with an internal consistency of $\alpha = .83$ (*Buysse et al., 1989*).
2. *Sleep disturbance in people living with dementia* measured *via* the SDI. The SDI consists of seven items from the Neuropsychiatric Inventory sleep disturbance subsection (*Cummings, 2020*). Caregivers are asked to rate the frequency, severity and levels of caregiver distress associated with sleep-related behaviours displayed by the person in their care.
3. *Insomnia* symptoms measured by the Insomnia Severity Index (*Bastien, Vallières & Morin, 2001*). The ISI is a 7-item measure of self-reported insomnia symptoms. The ISI has been shown to have high internal consistency and a cutoff score of $\geq 8$ has a sensitivity 96%–99% and specificity of 78%–92% in community and clinical populations respectively (*Morin et al., 2011*).
4. *Sleep-related daytime impairment* measured by PROMIS Sleep Related Impairment–Short Form–8a (PROMIS-SRI; *Yu et al., 2011*).
5. *Mood* measured by PROMIS Depression–Short Form–8a and PROMIS Anxiety–Short Form–8a (*Pilkonis et al., 2011*), both 8-item measures of depression and anxiety

**Table 4   Timing of measurements.**

| Timepoint | Enrolment<br>-t1 | Allocation<br>t0 | Post-allocation | | |
| --- | --- | --- | --- | --- | --- |
| | | | T1<br>Week 0 | T2<br>Week 6 | T3<br>Week 10 |
| Enrolment | X | | | | |
| Informed consent/assent Screening | X | | | | |
| Allocation | | X | | | |
| Intervention | | | | | |
| ASSESSMENTS | | | | | |
| Primary Outcome (Feasibility) | | | | | |
| Percentage of sessions attended by care-givers | | | | X | |
| Drop-out rate | | | | X | |
| Client Satisfaction Questionnaire | | | | X | |
| Secondary Outcomes (Preliminary Efficacy) | | | | | |
| Pittsburgh Sleep Quality Index | X | | X | X | X |
| Sleep Disturbance Inventory[a] | X | | X | X | X |
| Insomnia Severity Index | | | X | X | X |
| PROMIS Sleep Related Impairment –SF | | | X | X | X |
| PROMIS Anxiety –SF | | | X | X | X |
| PROMIS Depression –SF | | | X | X | X |
| Depression, Anxiety and Stress Scales | | | X | X | |
| Health Questionnaire (EQ-5D-5L)[a] | | | X | X | X |
| The Alzheimer's Disease-related Quality of Life scale[b] | | | X | X | X |
| Actigraphy[c]<br>• Sleep onset latency (mins)<br>• Sleep efficiency (%)<br>• Total sleep time (mins)<br>• Wake after sleep onset (mins) | | X | | X | |
| Other Factors | | | | | |
| Demographics | X | | X | X | X |
| Consensus Sleep Diary<br>• Sleep onset latency<br>• Sleep efficiency<br>• Total sleep time<br>• Number of overnight awakenings | | X | | X | |

**Table 4** (*continued*)

| Timepoint | Enrolment -t1 | Allocation t0 | Post-allocation T1 Week 0 | T2 Week 6 | T3 Week 10 |
|---|---|---|---|---|---|
| Reduced Morningness-Eveningness Questionnaire | | | X | X | X |
| PROMIS Emotional Support –SF –4a | | | X | X | X |
| PROMIS Instrumental Support –SF –4a | | | X | X | X |
| Intervention adherence and helpfulness | | | | X | X |
| Adverse events | | | | X | X |
| Credibility Expectancy Questionnaire | | | X | | |

**Notes.**
[T0] Pre-study screening/consent and baseline
[T1] Time-point 1 (Week 0) pre-intervention baseline questionnaire undertaken within one week of intervention start
[T2] time-point 2 (Week 6) post-intervention
[T3] Time-point 3 (Week 10) follow-up
[X] Undertaken at that time point
[SF] Short form
[a] Completed by caregivers only.
[b] Completed by persons living with dementia only.
[c] Actigraphy and sleep diary assessed for two weeks from pre-intervention to the start of Week 1, then from Week 5 to Week 6 (final two weeks of intervention).

symptoms respectively, and the Depression, Anxiety and Stress Scales (*Lovibond & Lovibond, 1995*).

6. *Quality of life* measured by the EQ-5D-5L (*Herdman et al., 2011*) for caregivers and the Alzheimer's Disease-related Quality of Life scale (*Rabins et al., 1999*) for people living with dementia.

7. *Objective sleep* measured *via* actigraphy. The actigraph (Actiwatch 2, Philips Respironics) is a small wrist-worn device resembling a wristwatch that incorporates an accelerometer, which detects body movement or lack of movement. Actigraphy data will be collected in conjunction with information from sleep diaries. An algorithm is then used to infer various sleep outcomes (*e.g.*, sleep onset, duration, time in bed) based on the absence of recorded movement. Although actigraphy does not measure sleep itself, the current algorithm of analyses made by these devices can provide feasible and valuable information about insomnia-related parameters and information related to circadian sleep-wake patterns that polysomnography is unable to provide (*Krystal & Edinger, 2008*). Actigraphy was verified using the Consensus Sleep Diary (*Carney et al., 2012*). Parameters used in the current study include: sleep onset latency in minutes, percentage of sleep efficiency, total sleep time in minutes, and wake after sleep onset in minutes.

## Other factors

1. *Demographic* information such as medical or mental health conditions, details on dementia diagnosis, living arrangements, medication use and engagement in other therapies.

2. Amount of daily *light exposure*, *daytime naps* and *physical activity* in minutes assessed through additional questions in the adapted sleep diary.

3. *Chronotype* assessed *via* Reduced Morningness-Eveningness Questionnaire (*Adan & Almirall, 1991*).

4. *Social support* measured using four item scales: PROMIS Instrumental Support–Short Form–4a and PROMIS Emotional Support–Short Form–4a (*Hahn et al., 2014*). PROMIS Instrumental Support has been used among caregivers and yields a high internal consistency of $\alpha = .90$ (*Kent et al., 2020*).

5. *Intervention adherence and helpfulness* will be assessed through rated frequency of use and helpfulness of various strategies (*e.g.*, practical sleep strategies, relaxation strategies, physical activity, guidance on light exposure, enlisting social support) through questions developed by the research team. Qualitative feedback will also be obtained.

6. *Adverse events* are assessed formally through questions developed by the research team.

7. *Credibility and expectancy of treatment* is assessed *via* the Credibility Expectancy Questionnaire (*Devilly & Borkovec, 2000*).

## Adverse events

Risks of participation are minimal and may include skin irritation caused by the actigraph band and psychological discomfort in completing questionnaires. Side-effects of participation will be formally assessed in the post-intervention and follow-up questionnaires. Any adverse events that occur throughout the trial will be recorded. Participants may discontinue actigraph use or completing questionnaires in the event of discomfort, and will be asked to communicate this to researchers. Participants will be encouraged to speak to their doctor in the event of ongoing poor sleep beyond completion of the trial and will be provided with details the following support services: National Dementia Helpline, Dementia Australia Counselling, Beyond Blue and Lifeline.

## Reimbursement

Participants will receive an $100 electronic gift voucher upon completion of the final questionnaire and return of actigraph and sleep diary. Participants are provided with a comprehensive list of relevant support services for ongoing support.

## Trial status

The project began recruiting in May 2022 and data collection is ongoing.

# DATA ANALYSIS

## Sample size

Given the feasibility nature of this study, a convenience sample of 24 dyads will be recruited; no power analysis was conducted.

## Data storage and management

All data will be stored under password protection. A tracking sheet contains personal details will be stored separately from responses to screening questions and questionnaires. Participants will be assigned a unique code to link responses across time-points. Interim data will only be accessible by investigators and research assistants involved in data collection and analysis; lead investigators and research assistants will have access to the

final dataset. Identifiable data will be kept for seven years following any publications and thereafter destroyed. To promote data quality, data will be screened for accuracy and further enquiries will be made by research assistants and rectified.

Participants may request further clarification of questionnaire items; help will be provided by research assistants *via* telephone/email. A data monitoring committee will not be employed due to limited scale, and auditing will not be undertaken. The trial will be terminated upon achievement of recruitment target.

### Statistical analysis

Descriptive data will be presented as means and standard deviation, or percentages where appropriate. Feasibility will be assessed *via* percentage of sessions attended, drop-out rate, and scores of 75% or more on the CSQ at post-intervention for caregivers. Changes in PSQI, ISI and SDI scores, actigraphy-assessed sleep (*i.e.,* sleep efficiency, sleep onset latency, wake after sleep onset, total sleep time), DASS, PROMIS Depression/Anxiety, EQ-5D-5L and Alzheimer's Disease-related Quality of Life Scale, and PROMIS Emotional/Instrumental Support scores from T1 to T3 will be assessed using linear mixed models, controlling for age and gender. Cohen's *d* will be used to estimate effect sizes.

### Dissemination

Results will be disseminated through peer-reviewed publications and conferences. Findings may also be disseminated through relevant organisations, support groups and media (including social media) outlets. Participants who elect to receive the results of the study will be contacted upon trial completion with a brief summary of findings. Those who have contributed significantly to the development, conduct, analysis and reporting of the trial will be granted authorship for any arising publications involving trial data. Authors may be contacted for requests on accessing materials, data and code.

## DISCUSSION

Over 91% community-dwelling people with dementia are supported by a caregiver at home (*Brooks, Ross & Beattie, 2015*), and in this population, sleep disturbances remain prevalent and are related to a range of undesirable outcomes. The prevalence of dementia is anticipated to rise steeply over coming decades, which will place significant demands on caregivers and dementia-related healthcare services (*Standfield, Comans & Scuffham, 2018*). Therefore, effective, feasible interventions that promote sleep health and wellbeing in caregivers are much needed to support people in their caregiving roles.

Existing sleep interventions commonly address sleep concerns in persons with dementia and caregivers separately (*Pignatiello et al., 2022*; *O'Caoimh et al., 2019*). Our intervention offers a novel approach by concurrently targeting sleep concerns in dyads of caregivers and care-recipients through multiple modalities in an attempt to more comprehensively and collaboratively address dyadic sleep disturbance. From a cognitive behavioural perspective, *perpetuating* factors of sleep disturbance are targeted, for example, caregivers may learn strategies to support their own healthy sleep practices and sleep health in the person they support which may contribute to their experience of sleep disturbance.

A strength of our trial was the engagement of lived experience perspectives in the design and development of the intervention. We engaged consumer and community perspectives (*i.e.*, through focus groups, interviews, questionnaires and iterative reviews of processes and materials from community advisors) in the development of this intervention and combined components from existing evidence-based interventions targeting caregivers and people living with dementia in an attempt to better meet the needs of the community. Additionally, the online and group-based delivery of the intervention aims to increase accessibility and timeliness of care to dyads experiencing poor sleep who may not otherwise be able to visit a clinician; thus improving health equity. This also allows for a much higher treatment frequency at a lower time cost, which may have a positive impact on individual compliance and ultimately on overall intervention success.

The feasibility nature of our trial will provide proof-of-concept data and limited-efficacy testing to support a larger-scale effectiveness-implementation trial. Results from our study will allow us to evaluate who may be better suited to this intervention (*e.g.*, people living with certain types of dementia, or dyads with certain types of sleep disturbance), as well as other considerations (*e.g.*, whether proxy or self-report assessments should be used) in future larger evaluation trials. Some limitations of the trial include: lack of specific assessment of dementia stage (*i.e.,* mild, moderate, severe) and subtype, which can contribute to nocturnal sleep disturbances; lack of assessment of whether participants completed past sleep studies and their results; online assessments which older adults may find challenging; and flexibility in questionnaire respondents (*i.e.,* some caregivers may complete questionnaires on behalf of the person they support which may risk bias).

If feasible and effective upon further testing, this intervention could represent an accessible, low-cost way of reducing sleep disturbance and negative daytime effects in dyads of co-residing caregivers and the people living with dementia in the community which may hold larger-scale societal and economic implications in the promotion of sleep and wellbeing in dementia and caregiving communities.

## ACKNOWLEDGEMENTS

The authors would like to acknowledge the Aboriginal and Torres Strait Islander peoples as the traditional custodians of the land upon which this trial has been developed and delivered, and wish to pay their respects to Elders past, present, and emerging. Tom Valenta and Ann Piestch drew upon their living experience in the design and development of this trial, for which the rest of the authoring team are hugely grateful. The people offered their knowledge, experience, guidance and suggestions to the research team. through focus groups and interviews had an enormous role in shaping this study for which the authors would like to express their appreciation and gratitude. The authors wish to extend their thanks to: Arthur (Chun) Leung for his assistance in data collection and Dr Claire Jenkins and Dr Susie Oh for facilitating intervention sessions, Dr Kathryn Ellis for providing clinical supervision, Kate Harding at Dementia Australia for guidance and suggestions particularly around in community engagement and undertaking dementia-friendly research; Tom Morris and consumer advisors at HammondCare. There is no off-label or investigational use in this study.

### Funding

This trial was funded by the Dementia Centre for Research Collaboration—Pilot Grant Scheme 2020. The development and delivery of focus groups was funded by an internal Turner Institute Sleep and Circadian Theme Consumer and Community Involvement Grant. The official trial sponsor is Monash University. The funding source and sponsor had no role in design of the current study and will have no role in undertaking of the trial, including data analyses, interpretation, or decision to submit results. The funders had no role in study design, data collection and analysis, decision to publish, or preparation of the manuscript.

### Grant Disclosures

The following grant information was disclosed by the authors:
Dementia Centre for Research Collaboration—Pilot Grant Scheme 2020.
An internal Turner Institute Sleep and Circadian Theme Consumer and Community Involvement Grant.
Monash University.

### Competing Interests

The authors declare there are no competing interests.

### Author Contributions

- Sumedha Verma conceived and designed the experiments, performed the experiments, prepared figures and/or tables, authored or reviewed drafts of the article, and approved the final draft.
- Prerna Varma conceived and designed the experiments, authored or reviewed drafts of the article, and approved the final draft.
- Aimee Brown conceived and designed the experiments, performed the experiments, authored or reviewed drafts of the article, and approved the final draft.
- Bei Bei conceived and designed the experiments, authored or reviewed drafts of the article, and approved the final draft.
- Rosemary Gibson conceived and designed the experiments, authored or reviewed drafts of the article, and approved the final draft.
- Tom Valenta conceived and designed the experiments, authored or reviewed drafts of the article, lived experience advisor (caregiver), and approved the final draft.
- Ann Pietsch conceived and designed the experiments, authored or reviewed drafts of the article, lived experience advisor (person living with dementia), and approved the final draft.
- Marina Cavuoto conceived and designed the experiments, authored or reviewed drafts of the article, and approved the final draft.
- Michael Woodward conceived and designed the experiments, authored or reviewed drafts of the article, and approved the final draft.
- Susan McCurry conceived and designed the experiments, authored or reviewed drafts of the article, and approved the final draft.
- Melinda L. Jackson conceived and designed the experiments, performed the experiments, prepared figures and/or tables, authored or reviewed drafts of the article, and approved the final draft.

## Human Ethics

The following information was supplied relating to ethical approvals (*i.e.,* approving body and any reference numbers):

Ethics approval was obtained by the Monash University Human Research Ethics Committee (30710). The trial was registered prospectively on the 27th of January 2022 with Australia and New Zealand Clinical Trials Registry (ACTRN12622000144718). Informed consent was obtained from all participants.

## Data Availability

There is no data available as this is a protocol. The sample documents (*i.e.,* sample intervention material, consent form) are available in the Supplementary Files.

## Supplemental Information

Supplemental information for this article can be found online at http://dx.doi.org/10.7717/peerj.16543#supplemental-information.

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
