# Peer review of "Multi-modal sleep intervention for community-dwelling people living with dementia and primary caregiver dyads with sleep disturbance: protocol of a single-arm feasibility trial"

_PeerJ, doi:10.7717/peerj.16543_

## Round 0.1 · original submission · Minor Revisions

Congratulations, the two reviewers and I are impressed with your study and the written manuscript that you have submitted to PeerJ. Please have a look at the constructive criticisms provided by the two reviewers prior to resubmitting the revised manuscript.

·

Basic reporting

Please, in supplementary materials, the frontiers logo should be removed.

I think that the theorethical framework of this intervention is unclear and could be improved.

Experimental design

Lines 263-269: I think the authors could briefly report some suggestions from participants in their member check validation. Furthermore, I think that the main goal of these focus groups/interviews should be reported.

If the intervention is personalized, how do authors think to report information related to replication and reproducibility?

Regarding tailoring, which variables differentiated and stratified the intensity of intervention?
Why authors decided to use them?

Please, provide detailed information regarding activities.

Authors should discuss the convenience sample based on previous literature.

Regarding statistical analysis why authors did not think about EMA?

Validity of the findings

In the discussion, the authors should focus on the novelty of the intervention.

Reviewer 2 ·

Basic reporting

This was a well-written paper that justifies the need for the study with a relevant research question. The methods were well described with a few areas needing further clarification. A few minor grammatical clarifications are also needed.

In the abstract and the manuscript, clarify the difference between therapist-assisted and therapist-led.
The sentences would be cleared if “supported by a workbook” was a new sentence.

Pg 9 line 117: The phrase “sleep quality and characteristics are multifactorial,” appears to be incomplete.

P11, Line 170-172: This sentence needs to be clarified “Given that in dyadic interventions, sleep disturbances in both the caregiver and the person they support are simultaneously addressed (22), which may increase reach to populations of caregivers and people living with dementia experiencing poor sleep.

Experimental design

Some clarification is needed regarding how the person living with dementia would be identified and included in the study. Would it be caregiver reported with a medical diagnosis? Or a chart review or based on medications prescribed? What is the process for screening the person living with dementia and the process for getting the consent/assent from the person living with dementia?

The process for data collection using the actigraphs is unclear. Will they be returned after the second week and then new ones mailed for the 5th and 6th weeks? Can the watch be programmed to only collect data during those specific times? Can the watch to be used remain charged for that length of time or will the charger be sent, and instructions given regarding how to charge the watch? Is the data uploaded daily so the research assistant can monitor whether they are wearing the watch? What happens if the watch isn’t working when the participants receive it?

Pg 16: What is meant by “The intervention will be discontinued following participant request.”

Pg 16, Ln 339-330: suggest replacing assess with “gather” and place “assess the” in front of preliminary.

The participants must complete and submit the data for all three time points before being reimbursed? For the future, perhaps consider reimbursement after each data collection point so that if participants drop-out before the final data collection point, they would have received reimbursement for the data points they completed.

Validity of the findings

No comment

Additional comments

Review references for completeness e.g. #41

---

## Round 0.2 · Minor Revisions

Thanks very much for your attendance to the constructive criticisms of the two reviewers. There are some very small details left for you to attend to based on the comments of reviewer two, before we can accept this submission for publication in PeerJ.

·

Basic reporting

This was a well-written paper that justified the need for the study with a relevant research question. The current version was updated and improved. Congratulations!

Experimental design

no comment

Validity of the findings

no comment

Additional comments

no comment

Reviewer 2 ·

Basic reporting

Few grammatical mistakes.
Pg 5, Lns 123-126: Please clarify "Sleep characteristics, such as sleep quality, in caregivers supporting people living with dementia are multifactorial." Do you mean causes of sleep characteristics such as poor sleep quality...?
Second sentence could be: Caregiver demographic and psychosocial factors, as well as sleep-wake behaviours in the person they support can all increase the risk for disturbed sleep.
Ln 228 – caregivers misspelled
Ln 229: delete with
Ln 265: Consider replacing "and" with "to be" in the statement "..will be posted and worn again during Weeks 5 and 6 of the intervention."

Experimental design

No comment

Validity of the findings

No comment

---

## Round 0.3 · accepted · Accept

Thanks for the remaining minor corrections. I am happy to recommend this be accepted for publication in PeerJ